# Wnt/β-Catenin-Pathway Alterations and Homologous Recombination Deficiency in Cholangiocarcinoma Cell Lines and Clinical Samples: Towards Specific Vulnerabilities

**DOI:** 10.3390/jpm12081270

**Published:** 2022-08-01

**Authors:** Alexander Scheiter, Frederik Hierl, Ingrid Winkel, Felix Keil, Margit Klier-Richter, Cédric Coulouarn, Florian Lüke, Arne Kandulski, Matthias Evert, Wolfgang Dietmaier, Diego F. Calvisi, Kirsten Utpatel

**Affiliations:** 1Institute of Pathology, University of Regensburg, 93053 Regensburg, Germany; frederik.hierl@stud.uni-regensburg.de (F.H.); Ingrid.Winkel@ukr.de (I.W.); Felix1.Keil@ukr.de (F.K.); Margit.Klier-Richter@ukr.de (M.K.-R.); matthias.evert@ukr.de (M.E.); wolfgang.dietmaier@klinik.uni-regensburg.de (W.D.); diego.calvisi@klinik.uni-regensburg.de (D.F.C.); kirsten.utpatel@klinik.uni-regensburg.de (K.U.); 2Bavarian Center for Cancer Research/BZKF, 91054 Bavaria, Germany; florian.lueke@klinik.uni-regensburg.de; 3INSERM, Université de Rennes, F-35042 Rennes, France; cedric.coulouarn@univ-rennes1.fr; 4Department of Internal Medicine III, University Hospital Regensburg, Hematology and Oncology, 93053 Regensburg, Germany; 5Division of Personalized Tumor Therapy, Fraunhofer Institute for Toxicology and Experimental Medicine, 93053 Regensburg, Germany; 6Department of Internal Medicine I, University Hospital Regensburg, 93053 Regensburg, Germany; arne.kandulski@klinik.uni-regensburg.de

**Keywords:** cholangiocarcinoma, β-catenin, WNT, APC, HRD, Olaparib, PARP, intraductal papillary neoplasm of the bile duct

## Abstract

Cholangiocarcinoma (CCA) features a dismal prognosis with limited treatment options. Genomic studies have unveiled several promising targets in this disease, including fibroblast growth factor receptor (FGFR) fusions and isocitrate dehydrogenase (IDH) mutations. To fully harness the potential of genomically informed therapies in CCA, it is necessary to thoroughly characterize the available model organisms, including cell lines. One parameter to investigate in CCA is homologous recombination deficiency (HRD). While mutations in homologous recombinational repair (HRR)-related genes have been detected, their predictive value remains undetermined. Using a targeted next-generation sequencing approach, we analyzed 12 human CCA cell lines and compared them to 62 CCA samples of the molecular tumor board cohort. The AmoyDx^®^ HRD Focus Panel was employed to determine corresponding genomic scar scores (GSS). Ten of twelve cell lines harbored alterations in common HRR-related genes, and five cell lines were HRD-positive, although this parameter did not correlate well with Olaparib sensitivity. Moreover, functionally relevant *APC* and *β-catenin* mutations were registered, which were also detected in 4/176 (2.3%) samples on a CCA microarray. Although rare, these alterations were exclusive to large duct type CCA with associated intraductal papillary neoplasms of the bile duct (IPNB) in 3 cases, pointing at a distinct form of cholangiocarcinogenesis with potential specific vulnerabilities.

## 1. Introduction

The term cholangiocarcinoma (CCA) comprises a heterogeneous group of malignancies originating along the biliary tree or from mature hepatocytes [1,2]. Despite this heterogeneity, palliative treatment is primarily based on platinum-based cytotoxic therapies with limited activity. Consequently, the median overall survival is below 11 months in advanced CCA [3], highlighting the necessity of patient-tailored treatment strategies, which take into account targetable molecular alterations. The efficacy of such therapeutic concepts has recently been demonstrated in CCA by the seminal FIGHT-202 study, which led to the approval of Pemigatinib for *FGFR2* fusion-positive pretreated patients by the Food and Drug Administration (FDA) in 2020 [4,5]. Moreover, Ivosidenib, in *IDH1*-mutated tumors, entered the market the following year, based on the ClarIDHy trial reporting significant benefits in overall survival [6]. Given this proof of concept, the hunt is on for targeted treatment strategies. For further investigations on predictive biomarkers, it is obligatory to comprehensively characterize the various model organisms, such as CCA cell lines.

One emerging biomarker in other cancer entities is homologous recombination deficiency (HRD). In conjunction with non-homologous end joining, homologous recombination repair is responsible for resolving DNA double-strand breaks in mammalian cells [7,8]. Defects in the latter pathway result in genomic instability, one of the hallmarks of carcinogenesis [9]. Mutations in homologous recombinational repair (HRR) related genes, which include but are not restricted to *BRCA1/2*, *ARID1A*, *ATM*, *ATRX*, *BAP1*, *BARD1*, *BLM*, *BRIP1*, *CHEK1/2*, *FANCA/C/D2/E/F/G/L*, *MRE11A*, *NBN*, *PALB2*, *RAD50*, *RAD51*, *RAD51B*, and/or *WRN* might be responsible for HRD [10]. These alterations, particularly in *BRCA1/2* genes, can potentiate the effect of ionizing radiation and platinum-based cytotoxic chemotherapy [11,12]. Furthermore, inhibitory compounds of Poly (ADP-ribose) polymerase (PARP), an enzyme involved in the repair of DNA single-strand breaks (ultimately resulting in double-strand breaks during the replication process), can potentiate the deleterious effect of HRD. This combined impairment of DNA damage repair can lead to a detrimental accumulation of genomic errors, i.e., a genomic scar, a phenomenon known as “synthetic lethality” [13,14]. Based on this principle, PARP inhibitors have been approved for pancreatic, breast, prostate, and ovarian cancer [15,16,17,18]. Concerning biliary cancer, to date, only a few encouraging case reports exist for Olaparib treatment on the ground of *BRCA1/2* gene mutations [19,20,21,22]. An analysis across cancer types for HRR-related gene alterations discovered that biliary tract cancers belonged to the most frequently mutated entities (28.9%; *n* = 343), while *BRCA1/2* only accounted for 3% here [10]. Since many of these variants are of unknown significance, their interpretation requires a functional readout, which can be achieved by evaluating the genomic scar score as a surrogate for HRD [23]. The QUADRA trial, which brought about the FDA approval of Niraparib for late-line treatment of high-grade serous carcinoma of the inner genital, serves as evidence for the clinical validity of HRD scoring [24]. Thus, the measurement of HRD scores in CCA cell lines and HRR-related gene mutation frequencies in CCA samples represented a core incentive for this article.

Furthermore, CCA can harbor numerous additional genetic alterations, representing potential treatment targets. For instance, MAPK signaling pathway activation via diverse mechanisms occurs in CCA, among which the most prevalent are *RAS* mutations, followed by *BRAF* and *MEK* mutations [25]. Although not confirmed in vitro, a clinical trial in CCA suggests a positive predictive value in patients suffering from a biliary tract or gallbladder cancer with *KRAS* or *NRAS* mutation [26].

Finally, a pathway extensively studied in cancer and potentially amenable to therapeutic interventions soon is the Wnt/β-catenin signaling [27,28]. In addition to colorectal carcinogenesis, where *APC* mutations initiate the adenoma-carcinoma sequence [29], activating *CTNNB1* (β-catenin) mutations occur in hepatocellular carcinoma and hepatoblastoma development. In contrast, such alterations are rare in CCA [30]. Nonetheless, robust activation of the Wnt/β-catenin pathway occurs in human CCA, most likely due to Wnt ligands produced by the tumor microenvironment [31]. Therefore, we analyzed CCA cell lines for Wnt/β-catenin signaling to demonstrate their suitability as models for this pathway and potential targeted inhibitors.

## 2. Materials and Methods

### 2.1. Cell Culture and In Vitro Analyses

The 12 human intrahepatic cholangiocarcinoma (iCCA) cell lines were cultured in 5% CO_2_ at 37 °C in a humidified incubator. SG231, TKKK CC-LP-1, KKU-M156, KKU-M213, KKU-100, and KKU-M055 were grown in Dulbecco’s modified Eagle medium (Gibco, Grand Island, NY, USA). HuCC-T1, RBE, and YSCCC were grown in RPMI 1640 medium (Gibco). OZ was cultured in William’s E Medium (Gibco) and HuCC-A1 in Ham’s F-12 Nutrient Mixture (Gibco). All media were supplemented with 5% fetal bovine serum (Gibco), 100 mg/mL streptomycin, and 100 U/mL penicillin.

Cell viability assays were performed using the xCELLigence^®^ real-time cell analysis dual plate (RTCA DP) device (OLS OMNI Life Science GmbH & Co KG; Bremen, Germany). For real-time impedance-based cell index measurement, cells were seeded on E-Plate 16 PET (Agilent Technologies, Inc., Santa Clara, CA, USA). Measurement sweeps were acquired every 15 min. Five to ten thousand cells, which were suspended in a volume of 150 µL of growth medium, were seeded in each well. After 24 h, varying concentrations of the cytotoxic compounds Olaparib (PARP-inhibitor) [range 3.13–200 µM] (MedChemExpress; LLC., Monmouth, NJ, USA), Ivosidenib (IDH1-inhibitor) [range 3.13–200 µM] (MedChemExpress), Foretinib (MET-inhibitor) [range 0.13–8 µM] (MedChemExpress), Trametinib (MEK-inhibitor) [range 3.13–200 µM] (Selleck Chemicals GmbH, Berlin, Germany) or Mirdametinib (MEK-inhibitor) [range 3.13–200 µM] (Selleck Chemicals GmbH, Planegg, Germany) were applied. The measurements were obtained in triplicates for a total of 72 h.

Raw data were analyzed with the RTCA software (OLS OMNI Life Science GmbH & Co., KG, Bremen, Germany). The data were normalized to the timepoint of inhibitor addition. Moreover, IC_50_ values were computed in the RTCA software using the formula for sigmoidal dose-response (variable slope) across the observational period.

### 2.2. Western Blot (WB)

Cell pellets were homogenized and lysed in T-PER Mammalian Protein Extraction Reagent (Thermo Fisher Scientific, Waltham, MA, USA) with the Halt^TM^ Protease and Phosphatase Inhibitor Cocktail (Thermo Fisher Scientific) at a ratio of 100:1. The Pierce BCA Protein Assay Kit (ThermoFisher Scientific) was used to determine the protein concentration. Protein denaturation was done in sample buffer (Laemmli Sample buffer: β-ME = 19:1) at 95 °C for 5 min. Proteins were then separated by sodium dodecyl sulfate-polyacrylamide gel electrophoresis. The gel was placed onto a nitrocellulose membrane (Bio-Rad Laboratories, Hercules, CA, USA). The membrane was blocked in EveryBlot Blocking Buffer (Bio-Rad Laboratories) at room temperature for 15 min and subsequently incubated with primary antibodies (Table 1) at 4 °C overnight. The next day, membranes were incubated with horseradish peroxidase-conjugated secondary antibodies (Table 1) at room temperature for 1 h. After washing with Tris Buffered Saline with Tween^®^ 20 (Cell Signaling Technology, Inc., Cambridge, UK), membranes were developed with the Clarity Western ECL Substrate (Bio-Rad Laboratories). Image acquisition was performed on a ChemiDoc^TM^ MP Imaging System (Bio-Rad Laboratories).

### 2.3. Immunohistochemistry

For immunohistochemical analyses, two µm thick histological sections were cut from formalin-fixed, paraffin-embedded (FFPE) tissue blocks. Sections were incubated for 30 min at a 70 °C temperature. Afterward, slides were deparaffinized through a series of xylene and gradient alcohols to water. For antigen retrieval, slides were cooked for 8 min at 110 °C in citrate pH 6.0 and then placed in iced water. Next, slides were incubated in 1× Dako Peroxidase-Blocking Solution^®^ (S2023; Dako GmbH, Jena, Germany) for 10 min. The primary antibody was diluted in Dako Antibody Diluent^®^ (S2022). Sections were incubated herein in a humidity chamber at room temperature overnight (see Table 2 for a list of employed antibodies). After washing with Dako washing solution^®^ (S3006), the secondary antibody Histofine Simple Stain MAX PO^®^ anti-rabbit or anti-goat was administered for 60 min at room temperature. Following two additional washes in Dako washing solution^®^ (S3006), the chromogenic reaction was performed with the Dako Liquid DAB + Substrate Chromogen System^®^. Slides were stained in Mayer’s hemalum for 10 s. Coverslips were automatically placed on top with the Ventana BenchMark Ultra^®^ (Roche, Basel, Switzerland). β-catenin antibodies used with respective dilutions are listed in Table 1.

### 2.4. Slide Image Aquisition

The slide scanner Pannoramic 1000 Flash RX^®^ (Sysmex Europe SE, Norderstedt, Germany) was used for image acquisition. The 20× magnification objective was selected. Stitched images were visualized using the SlideViewer^®^ (Sysmex) software. Screenshots of relevant regions were generated with a 300 ppi resolution.

### 2.5. Nucleic Acid Extraction and Quantitative Reverse Transcription Real-Time Polymerase Chain Reaction (qRT-PCR)

Nucleic acids from formalin-fixed, paraffin-embedded patient (FFPE) tissue were isolated using the Maxwell^®^ RSC RNA FFPE Kit (Promega GmbH, Walldorf, Germany) and Maxwell^®^ RSC (Promega) device. For RNA concentration, the Agilent TapeStation^®^ (Agilent, Santa Clara, CA, USA) and a fluorometer (Qubit; Thermo Fisher Scientific) were used.

The AllPrep^®^ DNA Mini Kit (Qiagen Sciences Inc., Germantown, TN, USA) was used for DNA extraction from CCA cell lines. Total RNA extraction from CCA cell lines was performed with the NucleoSpin^®^ RNA Plus Kit (Macherey-Nagel GmbH & Co, Düren, Germany). RNA was converted to cDNA using the High-Capacity cDNA Reverse Transcription Kit with RNase Inhibitor (Thermo Fisher Scientific). qRT-PCR was performed on the CFX96^TM^ Real-Time System (Bio-Rad Laboratories, Inc.). The following protocol was applied: 1.0 µL primer mix, 5 µL TaqMan Universal PCR Master Mix (ThermoFisher Scientific), 2.5 µL cDNA (100 ng), and 2.1 µL d2H_2_O. Cycling conditions: polymerase activation at 95 °C for 2 min, 40 cycles: denaturation at 94 °C for 30 s, annealing at 55 °C for 30 s, and extension at 72 °C for 30 s. Primers for qRT-PCR were ordered from Thermo Fisher Scientific and listed in Table 2.

### 2.6. Next-Generation DNA and RNA Sequencing

The hybrid capture-based TruSight Oncology 500 Library Preparation Kit (Illumina, San Diego, CA, USA) was employed for library preparation according to the manufacturer’s protocol. Unique molecular identifier (UMI)-containing adaptors were ligated to the DNA fragments. Next, a pool of oligos specific to the 523 genes included in TSO500 were hybridized to the DNA libraries. Streptavidin magnetic beads captured the probes hybridized to the targeted regions. Primers were used to amplify the enriched libraries before purification with sample purification beads. Afterward, the libraries were pooled, denatured, and diluted. TSO500 libraries were sequenced on a NextSeq^TM^ 550Dx (Illumina). Variant annotation was performed using Qiagen Clinical Insight (QCI) Interpret (Qiagen N.V., Venlo, The Netherlands). In addition, the software vcf2maf converter (https://github.com/mskcc/vcf2maf; accessed 1 June 2022) was employed to generate maf files, which were uploaded to a locally installed version of cBioportal [32,33] for analysis of patient and cell line data.

### 2.7. HRD Score Analysis

Genomic DNA from CCA cell line samples was extracted for library construction and analyzed with the AmoyDx^®^ HRD-focus panel (Amoy Diagnostics, Shanghai, China) to screen 24,000 single nucleotide polymorphisms for HRD scoring [23]. 100 ng of DNA were applied from each sample. Sequencing was performed on a NextSeq^TM^550Dx device (Illumina). The calculation of the HRD score (GScore) was done by summing loss of heterozygosity, large-scale state transition, and telomeric allelic imbalance by means of the ANDAS software (Amoy Diagnostics). HRD-positivity was defined as a GScore ≥ 50 following the manufacturer’s instructions.

### 2.8. Nanostring and Statistical Methods

The comparison of differential mRNA expression was carried out using the NanoString nCounter (NanoString Technologies, Inc., Seattle, WA, USA). This technology relies on direct molecular barcoding of defined target mRNA sequences. Specifically, 100 ng of total RNA (OD260/280 ratio 1.7–2.2) was hybridized overnight with reporter and capture code set at 65 °C. Excess probes were washed off using a two-step magnetic bead-based purification on the nCounter Prep station (NanoString Technologies). The purified complexes were eluted off the beads, immobilized on a cartridge, and subsequently aligned. Data were collected using the nCounter Analyzer (NanoString Technologies) at 555 fields of view through epifluorescence microscopy and CCD capturing. The nCounter^®^ PanCancer Pathways panel (NanoString Technologies), including 770 genes from thirteen canonical pathways, was chosen.

Data were analyzed using the nSolver^TM^ analysis software Version 4.0 (NanoString Technologies). All samples passed quality control in this software. Additional analyses were conducted using the nCounter Advanced Analysis 2.0 plug-in (NanoString Technologies). Automated normalization was chosen by selecting those genes that minimize the pairwise variation statistic. For visualization, unsupervised clustering was selected to generate a heatmap based on the normalized data counts of individual mRNAs. Differential expression was plotted as a volcano plot with individual genes −log10. (*p*-value) and log2 fold change compared to the control group. The Pathview module was used to assess gene expression changes within the WNT pathway visually. The undirected global significance score of the included Kyoto Encyclopedia of Genes and Genomes (Kegg) gene sets was calculated and displayed as a heatmap to determine group pathway changes. Details can be obtained from the nCounter Advanced Analysis (Version 2.0.; NanoString Technologies®, Inc., Seattle, WA, USA) documentation. For differential expression analyses, *p*-value adjustments with the Benjamini-Yekutieli method were selected.

### 2.9. Patient Samples

All patients provided written informed consent under our institutional review board-approved protocol, and the study was conducted following the Declaration of Helsinki. FFPE tissue specimens of 62 CCA samples of the molecular tumor board cohort (34 intrahepatic cholangiocarcinomas (iCCA), 10 perihilar cholangiocarcinomas (pCCA), 9 distal cholangiocarcinomas (dCCA), 1 mixed hepatocellular cholangiocarcinoma (HCC/CCA), and 8 gallbladder carcinomas) and 4 CCA samples of a tissue microarray (among a total of 176 iCCA samples) displaying positive nuclear β-catenin staining were analyzed using the TSO500^®^-targeted sequencing panel. Tissue microarray samples were obtained from the primary hepatic tumors. The tissue samples of our molecular tumor board registry were derived either from the primary tumor (39/62) or from metastatic sites (23/62).

## 3. Results

### 3.1. Genetic Analysis of CCA Cell Lines in Comparison with Human CCA Samples

RNA and DNA extracts of the twelve iCCA cell lines under investigation were subjected to TSO500^®^-based targeted sequencing of 523 cancer-related genes. These analyses met the prespecified quality control parameters of coverage in exonic nucleotides of a minimum of 50 UMI-reads of 90%. In addition, the variant call cut-off was set to a 5% allele fraction. Panel-based microsatellite analysis was possible for all samples, since a sufficient sequencing depth was achieved for more than 40 monomorphic microsatellite loci in each case. Moreover, an evaluation of the tumor mutational burden (TMB) was performed, as the sequenced region was more than 1 MB with a sufficient sequencing depth. RNA-based analysis for genetic rearrangements and a splice variant analysis were also possible for all studied cell lines. The same analyses were carried out on 62 CCA patient samples to compare the frequencies of the detected genetic events and assess the available CCA cell lines as a suitable disease model.

Based on these results, we first evaluated the occurrence and frequencies of putative driver mutations as annotated by either OncoKB (Figure 1a and Appendix A) or QCI (Table 3). The distribution of the most commonly mutated genes was highly similar between the 12 CCA samples and the patient cohort of CCAs despite differences in the frequency, so the available CCA cell lines mirror well the heterogeneity in patients’ samples. Oncogenic *KRAS* mutations occurred in 58% of cell lines as opposed to 16% of patients. In patients, *KRAS* mutations were detected more frequently in pCCA (40%, 4/10) and dCCA (44.4%, 4/9) than in iCCA (5.9%, 2/34), while none was found in the gallbladder carcinoma specimens (0%, 0/8). *TP53* tumor suppressor gene mutations were observed in cell lines and CCA tissue samples with 50 and 23% occurrence frequencies, respectively. Furthermore, tumor suppressor genes with a mutational frequency of >10% in cell lines involved *SMAD4*, *SMARCA4*, *ARID1A*, *ARID2*, and *RBM10*. Notably, the latter 4 genes are either implicated in chromatin remodeling or heterochromatin assembly in the case of RBM10 [34,35]. *IDH1* mutations were found in 2 cell lines (17%, 2/12) and 4 CCA samples (7%, 4/62). A *TERT*-promoter mutation was exclusive to one cell line. Individual mutations in cell lines were registered in the histone methyl transferases *KMT2A* and *DNMT3A*, the former also in one CCA sample. *NSD1*, a ligand-regulated transcription factor activated by steroid hormones, while also possessing activity as a histone methyl transferase [36], was mutated in KKU-M213 cells. Interestingly, one gain of function oncogenic mutation in the *MAP2K1* gene could be found in a CCA cell line (p.K57T, negative regulatory domain) and in a case of gallbladder adenocarcinoma (p.G128D, kinase domain). The former mutation has been described to confer sensitivity to a combination of trametinib and panitumumab in a case of colorectal cancer before [37], while the latter mutation has been associated with resistance to MEK inhibitors [38]. *TSC1*, *FBXW7*, and *SPEN* were each mutated in one cell line and one CCA sample. Especially the role of the ubiquitin ligase FBXW7 has been implicated in cholangiocarcinogenesis via activated AKT signaling and the induction of c-Myc [39]. Of note, the detected mutations in *APC* and *CTNNB1* in CCA cell lines further suggest an involvement of the WNT pathway, which will be extensively discussed below. Also, considering variants of unknown significance, mutational frequencies of genes defined in various Kegg pathways were analyzed (Appendix A). Mutations could preferentially be detected in genes of the receptor tyrosine kinase (RTK)-RAS pathway, again with most mutations found in *KRAS*. Interestingly, CCA cell lines harbored mutations with unknown significance in *ERBB2* (50%) and *MAP2K2* (25%) with a high frequency. *MAP2K2* mutations could, however, not be found in the CCA patient cohort. Here also, various alterations could be detected in the RTK-RAS pathway, including *ERBB2* (8.2%), *ERBB3* (8.2%), *FGFR2* (16.4%), *MET* (9.8%), *NRAS* (4.9%), and *MAP2K1* (3.3%) as potentially druggable targets. Furthermore, mutations occurred in relevant genes of several more pathways both in CCA cell lines and tissue samples: the PI3K pathway (e.g., *PIK3CA*, *TSC1* and *2*, *AKT1*, and *MTOR*), the Notch pathway (*NOTCH1-4*), which has been proven to be a prerequisite for cholangiocarcinogenesis [40,41], and the cell cycle/ apoptosis regulating pathway (*RB1*, *FBXW7*, *TP53*).

Next, the GScore was evaluated in CCA cell lines and correlated with mutations in HRR-related genes ARID1A, ATM, ATRX, BAP1, BARD1, BLM, BRCA1, BRCA2, BRIP1, CHEK2, CHEK1, FANCA, FANCC, FANCD2, FANCE, FANCF, FANCG, FANCL, MRE11, NBN, PALB2, RAD50, RAD51, RAD51B, and RAD51D (Figure 1b, Appendix A). Among these genes, putative driver mutations were detected in RAD50 (K722Rfs*14 in HuCC-T1) and RAD51D (K111Ifs*13 in YSCCC). Of note, HuCC-T1 was determined to be HRD-positive with a GScore of 85.7, while the frameshift mutation in RAD51D in YSCCC did not result in high HRD (GScore of 40.3). Moreover, in the CCA cell lines, two putative driver mutations were found in ARID1A (P946Lfs*20 in KKU-M055 and S1138* in HuCC-A1 cells, respectively). In the case of P946Lfs*20, a high GScore of 57.4 was detected in KKU-M055 cells, indicating homologous DNA repair deficiency. In contrast, the S1138* stop gain mutation in HuCC-A1 was associated with a very low GScore of 7.7. Extending the analysis to variants of unknown significance, 10/12 cell lines (83.3%) harbored mutations in HRR-related genes. Altogether, 5/12 cell lines (41.6%) were considered HRD-positive by the AmoyDx HRD Focus Assay. A correlation between HRR-related genes and HRD-positivity was not evident. Concerning the frequency of individual HRR-related gene mutations in CCA cell lines, ARID1A mutations were most prevalent (5/12, 42%), followed by BRCA2, CHEK2, BARD1 (2/12, 8% each), and BAP1, BRCA1, BRIP1, FANCA, FANCD2, FANCE, FANCF, PALB2, RAD50 and RAD51B and RAD51D (1/12, 8% each). Similarly, in CCA tissue samples, the order was as follows: ARID1A (16/62, 25%), ATM (11/62, 17%), BAP1 (7/62, 11%), BRCA2 (5/62, 11%), BARD1 (5/62, 8%), and several more below 8% frequency. In total, in 45/62 (72.5%) of CCA samples, pathogenic mutations or variants of unknown significance in HRR-related genes were found, while 24/62 (38%) of samples harbored pathogenic mutations.

Apart from HRD, 2/12 (16%) of CCA cell lines had a TMB > 10 mut/MB (YSCCC cells with 34.6 mut/MB and KKU-M213 cells with 46.1 mut/MB), while a high TMB was only detected in 1 iCCA among 43 evaluable tissue samples (2.3%). Microsatellite instability (MSI), as inferred from panel diagnostics and defined as >10% unstable MSI sites, was not present in the cell lines. One of 25 MSI (4%) evaluable patients among the CCA tissue samples reached the criteria for MSI, as evaluated by the targeted panel diagnostics.

In the RNA analysis, two gene fusions were found: MET-CAV1 in CC-LP-1 cells and EWSR1-ADRBK2 in SG231 cells (Table 3). The *MET* breakpoint in the former fusion was located in exon1 so that no functional kinase domain would result. The functional significance of the detected EWSR1-ADRBK2 fusion is unclear. *EWSR1* fusions have rarely been described in adenocarcinomas such as colorectal cancer. The underlying mechanism is usually an enhanced activation of a C-terminal transcription factor [42]. In this case, the fusion partner encodes for G protein-coupled receptor kinase 3, so this classical mechanism is not fathomable here. In the *MET*-splice analysis, several different splice variants were detected in 5/12 (42%) of CCA cell lines. In part, multiple splice variants were detectable in a single cell line, such as OZ and KKU-M156 (Appendix A). However, no classical *MET*-Exon14-Skipping alteration was present.

Finally, we analyzed copy number variations (CNV) of the 59 genes included in the TSO500^®^ panel. CNVs ≥ 2 were considered relevant (Table 3). Primarily interesting was the discovery of a high CNV of *ERBB2* (10.840) in TKKK cells and two cases of EGFR CNVs > 2. Both alterations can represent important oncogenic drivers and are potentially amenable to targeted therapies. Likewise, the CCA samples also included one *EGFR*-CNV positive iCCA (1/62, 1.6%; CNV 13.582), which was confirmed by immunohistochemistry (H-Score 270), and one *ERBB2*-CNV positive extrahepatic CCA (CNV 11.896). In addition, CCA cell lines also displayed *RICTOR* copy number gains (3/12, 25%), which were recapitulated in human CCA samples, where 3/62 samples (4.8%) were positive for *RICTOR* copy number gains (which were, however, <2 with uncertain relevance).

### 3.2. Functional Characterization of CCA Cell Lines Concerning Genomic Alterations

To determine the biological relevance of the detected genetic alterations in CCA cell lines, Western Blot analyses were performed (Figure 2). In the case of *KRAS* driver mutations, an elevated ratio of phosphorylated ERK/ total ERK was evident in HUCC-T1, KKU-M055, KKU-100, KKU-M213, OZ, and SG231 cell lines. In contrast, the cell lines KKU-M156 and RBE only displayed a minor elevation in the ratio compared to KRAS WT cell lines CC-LP-1 and HuCC-A1, even though they harbored classical activating *KRAS* mutations at positions G12 and G13. Particularly noteworthy is that the KKU-M156 cell line, derived initially from KKU-M213 cells, is not defined by strong ERK activation, although the mutational background is highly congruent, and the same *KRAS* G13C mutation is present in both cases. KKU-M055 containing a *MAP2K1*-mutation showed a slightly elevated ERK ratio. The cell line TKKK also showed a high ratio of phosphorylated ERK despite being *KRAS* WT, presumably because of *ERBB2* amplification The described *EGFR* copy number gains were confirmed to result in an elevated expression of EGFR in the case of HuCC-A1 and YSCCC cells, while elevated protein levels were not observable in TKKK. In addition, the *ERBB2*-copy number gain in TKKK translated into a high protein expression. The diverse reported *MET* splice variants were not clearly associated with a change in c-Met protein levels. Regarding RICTOR copy number gains, low to medium Rictor protein levels were observed by Western Blotting. The effectors of mTORC2 signaling p-EPHA2 and p-GSK3β were in part upregulated (in CC-LP-1 and HuCC-A1). In addition, p-RPS6 and p-4EBP1, canonical effectors of mTORC1, were positive in most cell lines.

Next, we investigated the predictive value of variants detected in the CCA cell lines (Figure 3, Table 4). First, we analyzed a potential correlation between GScore/HRD and sensitivity to the PARP inhibitor Olaparib. IC_50_ values were calculated for a total of 10 cell lines, which were suitable for xCelligence^®^ gradient concentration experiments (OZ and TKKK were omitted because of very slow growth characteristics). The two cell lines with the highest GScore (>80) were found to be most sensitive to Olaparib treatment and had IC_50_ values of 59 and 91 µM, respectively (KKU-M156 and HuCC-T1). However, two other cell lines evaluated as HRD-positive demonstrated IC_50_ values that were either in the range of HRD-negative cell lines (KKU-M213, 217 µM) or even much higher (KKU-M055, 4150 µM). When comparing the IC_50_ values for the MEK-inhibitors Mirdametinib and Trametinib with regard to *KRAS* mutational status, no difference could be observed. Using the MET-inhibitor Foretinib, the CC-LP-1 cell line, which contained a MET-CAV1 fusion and several MET splice site variants of unknown significance, had the lowest IC_50_ value (1.05 µM) compared to cell lines without *MET* splice variants. Surprisingly, the two *IDH1* mutated cell lines RBE and YSCCC did not show a decreased IC_50_ compared to *IDH1* WT cell lines when using the IDH1 inhibitor Ivosidenib. Possible confounders could be the cooccurrence of a *KRAS* mutation in RBE cells, an unusual event that has only been described in one case of biliary cancer to date [43], and the high copy number gain of *EGFR* in YSCCC cells. Both of these concomitant alterations could inhibit the antiproliferative effects of Ivosidenib.

### 3.3. Genetic Alterations in the WNT-Pathway Are Rare in CCA but Define a Unique Type of Cholangiocarcinogenesis

Given the surprising occurrence of a pathogenic *CTNNB1* mutation (p.G34V) in CC-LP-1 cells and a likely pathogenic *APC* mutation in the KKU-M055 cell line (p.D1636fs*2) (Table 3), we broadened our analysis of the WNT pathway to further associated genes- including variants of unknown significance- as defined by the Kegg pathway. Compellingly, a considerable frequency of WNT pathway alterations could be found (Figure 4a). Specifically, 3.3% (2/62) of CCA tissue samples were positive for mutations in *CTNNB1* (Y30*/pCCA and I610M/ iCCA). Furthermore, besides the *CTNNB1*-mutation, CC-LP-1 also harbored an *APC* variant of unknown significance (A1670V). Hence, the frequency of *APC* alterations in CCA cell lines was 16.7% (2/12) compared to 6.6% of CCA samples (4/62, E1317Q/iCCA, R653=/iCCA, R99W/ iCCA, G2171W/iCCA). Furthermore, alterations could be found in Ring finger protein 43 (RNF43) in 1 CCA cell line (R117H, SG231) and 2 CCA tissue samples (R337Q/iCCA and E187*/gallbladder adenocarcinoma). RNF43 is a negative regulator of WNT/β-catenin signaling by reducing the membrane level of Frizzled [44]. Finally, *AMER1* displayed alterations both in CCA cell lines and CCA tissue samples, while mutations in *AXIN1*, *AXIN2*, and *TCF7L2* were exclusive to patients’ samples.

To evaluate the effect of the observed (likely) pathogenic mutations in *CTNNB1* and *APC* in the two CCA cell lines CC-LP-1 and KKU-M055, Nanostring^®^ gene expression analyses using the PanCancer Pathway Panel were conducted in comparison with the two WNT pathway WT cell lines HuCC-A1 and TKKK (Figure 4b,c). Here, a principal finding was the strong and highly significant upregulation of *AXIN2* (10.9 log2 fold change, adj *p* = 3.71 × 10^−18^; Appendix A) in both mutated cell lines, which we verified by qRT-PCR and Western Blot analysis (Figure 5). Since *AXIN2* is a primary target of canonical WNT signaling, its excessive activation clearly argues for the activation of the Wnt/β-Catenin/Tcf axis [45]. Simultaneously, *AXIN2* constitutes a principal negative regulator of the pathway, thus forming a negative feedback mechanism. This could explain the observed regulation of additional WNT pathway genes. For example, the WNT genes encoding Wnt ligands, WNT7A and WNT10A, are downregulated. Likewise, *CCND3,* another effector of the WNT canonical signaling, and *TBL1XR1*, which recruits β-catenin to its target genes, were significantly reduced [46]. In contrast, overexpression was observed for *CCND1*, another WNT canonical target, and *FZD7*, a receptor for WNT proteins and the Wnt ligand *WNT5A*. *MYC*, another canonical target of the WNT signaling, was also upregulated, albeit not statistically significant (*p* = 0.196; Appendix A). Canonical HCC and hepatocyte-specific WNT-target genes, including TBX3, GLUL, and OAT, were not differentially expressed in CC-LP-1 and KKU-M055 cells as determined by RT-qPCR (Figure 5a).

Nanostring heatmaps showing how pathway scores (fit using the first principal component of each gene set’s data) change across samples were generated using the nSolver^®^ software (Figure 4c). Here, deregulation of WNT-pathway scores could be demonstrated for CC-LP-1 and KKU-M055 cells, which clustered together as opposed to WT cell lines TKKK and HuCC-A1. Concomitant deregulation of TGFβ-signaling was also observed. These two pathways have been reported to cooperate in colorectal carcinogenesis [47]. Moreover, several additional pathways were jointly deregulated in *APC*/*CTNNB1* mutant cell lines, including Notch-, PI3K, and MAPK cascades. Looking at a visual representation of the pathways against a baseline of either CC-LP-1 or KKU55 cells, activation of WNT canonical targets can be seen compared to TKKK (Appendix A). The relatively low ratio of non-phosphorylated/total β-Catenin indicated another counterregulatory mechanism, which is the increased phosphorylation/inactivation of β-catenin. Moreover, total protein levels of β-catenin were comparatively low in the *APC*-mutated cell line KKU-M055 (Figure 5b,c). Altogether, the transcriptional pattern does not allow a clear distinction between WNT pathway activation and inhibition, and several counteracting mechanisms appear to be at play, such as the upregulation of *AXIN2*.

Next, we carried out β-catenin immunohistochemistry on a tissue microarray of 176 primary CCAs to search for patient samples mirroring the alterations found in the cell lines. Specifically, 4/176 (2.3%) displayed marked nuclear positivity of β-catenin tested with three different β-catenin antibodies (Figure 6), which can serve as a surrogate for the presence of activating *CTNNB1* mutations. Whole slides were also stained with equivalent results. By targeted sequencing using the TSO500^®^-panel, three patients were identified to harbor likely pathogenic *APC* mutations (patients (P)1–3), and one patient had a pathogenic activating β-catenin mutation (P4; Appendix A), explaining the increased nuclear translocation of β-catenin. Interestingly, all tumors were classified as large duct type iCCA, and 3 of 4 tumors had associated intraductal papillary neoplasms of the bile duct (IPNB), which have been shown to harbor mutations in *APC* and *CTNNB1* in a few cases before [48]. Another compelling finding was that P1 had metachronous colon cancer 4 years before the CCA, which was also characterized by nuclear accumulation of β-catenin (Appendix A). P2 had also developed colon cancer before, pointing at a predilection caused by the presence of *APC* mutations, albeit the germline status could not be evaluated.

To study the functional implications of the detected mutations, a gene expression Nanostring^®^-analysis was conducted. For comparison, 3 additional samples from our CCA sequencing cohort without *APC*/*CTNNB1* mutation were randomly selected (P5–7), and membranous β-catenin staining was verified by immunohistochemistry (Appendix A).

Similar results were obtained as in the comparison of CCA cell lines with regard to the WNT-β-Catenin pathway (Figure 7, Appendix A). Again, *AXIN2* was highly upregulated in all mutated CCA samples, indicating activation of the canonical *WNT* signaling and a negative feedback mechanism. As in the cell lines, *RAC2*, *CCND3*, *TBL1XR1,* and *RAC1* genes were significantly downregulated, and *CCND1* upregulated. As an additional finding to the cell line analysis, *DKK1*—a principal antagonist of the WNT signaling [49]—was strongly downregulated in mutated patients’ samples. *BAMBI*, which promotes nuclear translocation and WNT activation, was, in turn, upregulated [50]. Assessing the pathway score, again strong and homogenous deregulation of WNT signaling was evident (Figure 7b), while the direction of PI3K and TGFβ signaling appeared opposite to cell lines.

## 4. Discussion

This study aimed at comprehensive molecular profiling of CCA cell lines and comparison with actual patients’ tumor samples. Altogether, a frequency of major genetic alterations in accordance with the available literature could be determined [25]. Recurrent mutations included the tumor suppressors *TP53*, *SMAD4*, and the epigenetic modifying genes *ARID1A*, *ARID2*, *SMARCA4,* and *BAP1* were found in both cell lines and tissue samples. In terms of activating oncogenic mutations, *KRAS* and *IDH1* mutations were the most frequently observed. In addition, an *ERBB2* amplification was found in the cell line TKKK, which was also evident on the protein level. A compelling finding was the occurrence of *EGFR* CNVs in 3 cell lines, out of which two could also be verified on the protein level. In the case of HuCC-A1 cells, no other major oncogenic alteration was detected in addition to the *EGFR* copy number gain, so it might be assumed to play an essential role in the tumor biology in this case. However, this association is less clear in YSCCC, where an additional oncogenic *IDH1* mutation was found. Finally, in TKKK, the *EGFR* amplification could not be confirmed on the protein level, while the *ERBB2* amplification clearly prevailed in tumor biology in this case. *EGFR* amplifications could potentially represent a treatment target in CCA, which has been studied in a clinical trial with gemcitabine and oxaliplatin combined with the EGFR inhibitor panitumumab. Here, an increase in progression-free survival (PFS) and overall survival (OS) has been observed in the case of fluorescent in situ hybridization positive extrahepatic cholangiocarcinomas (eCCA), comprising both pCCA and dCCA, which was, however, not statistically significant. The association was instead not found for iCCA and appeared even reversed in gallbladder carcinoma [51], prompting further research in this area. Another interesting alteration detected in the MAPK pathway, which in total harbored many genetic mutations (also including VUS) and could thus be considered a major driver in cholangiocarcinogenesis, was a *MAP2K1* mutation in the KKU-M055 cell line. This mutation also resulted in an increase in the p-ERK/ERK ratio. Nonetheless, this increase was lower than most *KRAS* mutations, which overall showed the most substantial effect on this parameter. In the CCA cohort, another driver *MAP2K1* mutation occurred in a case of gallbladder adenocarcinoma (p.G128D). *MAP2K1* activating mutations are known in the literature to play a critical role in histiocytic neoplasms, where in individual cases, responses to a cobimetinib therapy have been described [52]. Furthermore, a phase I study of binimetinib in patients with non-small cell lung cancer, among whom 8 harbored MAP2K1 mutations, reported an overall response rate of 42.8% [53]. Thus, despite being exceedingly rare, these mutations could constitute a treatment target in CCA and should be considered potentially actionable in molecular tumor boards. Apart from alterations in the MAPK pathway, mutations were also detected in the Notch pathway and the PI3K pathway, which could also be therapeutically harnessed. Importantly, the pharmacological analysis of the MEK inhibitors Mirdametinib and Trametinib did not reveal *KRAS* mutations as a positive predictive factor, which is contrary to retrospectively analyzed findings of a study of binimetinib plus capecitabine in CCA patients, where RAS/RAF/MEK/ERK pathway mutations showed significantly better tumor response, PFS and OS [26].

Altogether, cell lines had a higher proportion of oncogenes and tumor suppressor genes, while maintaining similar mutational characteristics to patients’ samples. The increased occurrence of oncogenes could favor the growth characteristics of CCA cell lines, increasing their aptitude for research. Thereby, a slight bias is incorporated into experiments involving CCA cell lines. This should be accounted for when choosing in vitro models to mirror the human disease.

Next, a comprehensive analysis of HRD in CCA cell lines was performed. The finding that a large percentage of CCA cell lines (41.6%, 5/12) were HRD positive was unexpected, especially since no clear association to known driver mutations in DNA-damage related (DDR) genes could be established. *ARID1A* mutations were found not to be a good discriminator for HRD positivity, as they were found both in HRD positive and negative CCA cell lines. In contrast, a driver mutation observed in *RAD50* in HuCCT-1 occurred together with a high GScore, presumably indicating an association. When extending the analysis to VUS of DDR genes, 10/12 cell lines harbored alterations in DDR genes, and overall the affected genes were similar between cell lines and patients’ samples. Of note, 38% (24/62) of patients’ samples harbored pathogenic mutations in HRR-related genes. Although the association to HRD might not be clear, the frequency should warrant further experimental elucidation to determine the extent of HRD positivity in CCA and specify the relevant HRR-related genes and mutations that result in a clinically actionable phenotype. Undoubtedly, the high frequency in CCA cell lines is an encouraging finding in this respect. Since individual studies have purported the predictive value of *BRCA1/2* mutations in CCA [20], it would be highly desirable to increase the group of patients who could benefit from PARP inhibition. Initial attempts to demonstrate a correlation between the HRD score and Olaparib sensitivity were made in the present study. The two cell lines with the highest HRD score were most sensitive to PARP inhibition. However, in a few instances, GScore high/HRD positive cell lines did not respond significantly better to PARP inhibition than GScore low/HRD negative cell lines. This might indicate that the cut-off is not yet well determined and should potentially be set higher than the arbitrary value of 50 currently proposed by the manufacturer. Importantly, in this study of cell lines, the two *IDH1* mutations did not result in an increased GScore, although IDH1 plays a critical role in DNA repair (with α-ketoglutarate being essential for the function of the DNA repair enzyme ALKBH [43]). The clinical translation of PARP inhibition in CCA is already underway [54]. It will be paramount to focus the analysis on HRD scores and DDR gene-related defects, given the low correlation we report here for CCA cell lines.

Finally, we set out to study the importance of the WNT/β-catenin pathway prompted by the occurrence of an *APC* and a *β-catenin* mutation in cell lines. In CCA, the activation of the β-catenin pathway has been evaluated by immunohistochemistry before. In particular, one study reported a reduction in membranous expression in 82% (of 71 samples) and an erroneous nuclear expression in 15% of samples. However, the reason for the nuclear localization of β-catenin had not been determined in this particular study, potentially because only *CTNNB1* mutations were analyzed, which could not be found there [55]. In contrast, in this study, we found a total of 4 out of 176 CCA samples with aberrant nuclear expression by β-catenin immunohistochemistry. Here, using next-generation sequencing, the cause for the aberrant nuclear expression could be attributed to either *CTNNB1* or *APC* mutations. Of particular interest is the finding that the respective CCAs with aberrant nuclear expression were all CCA of the large duct type. On closer examination, an association with IPNB could be determined in three of four cases, clearly delineating a special mode of cholangiocarcinogenesis occurring in the large bile ducts and involving the classical canonical WNT pathway. In contrast, the non-canonical WNT pathway has been postulated to play a critical role in intrahepatic cholangiocarcinogenesis [31]. Of note, two of 4 WNT pathway mutation positive CCAs had been preceded by colorectal cancer. One of these colorectal cancers could be demonstrated also to display an aberrant nuclear β-catenin localization. Thus, it could be assumed that in these patients, a specific predilection for colorectal cancer and CCA exists. By transcriptomic analyses of both cell lines and CCA samples, the activation of the canonical WNT signaling could be verified, especially by the remarkable upregulation of *AXIN2*. The occurrence of β-catenin in IPNB has been described before, albeit without the association with invasive carcinoma [56]. The gap to invasion could clearly be closed by the present study, as aberrant nuclear β-catenin expression was present both in the precursor lesions and adjacent invasive carcinomas. Although this mode of cholangiocarcinogenesis appears to be rare, it is imperative to know the WNT-pathway mutational status to study the effect of WNT inhibitors and select suitable patient cohorts. A pivotal experimental study by Boulter et al. [57] determined the cell line CC-LP-1 initially to be highly sensitive to the inhibitor Foscenvivint (ICG-001), which antagonizes Wnt/β-catenin/TCF-mediated transcription and binds specifically to CREB-binding protein. In retrospect, this sensitivity could well be attributed to the *CTNNB1* mutation detected here. This finding hints at the ability to harness WNT pathway alterations therapeutically, and large duct type CCA originating preferentially from IPNBs could be potential candidates for such treatments.

Overall, the current study provides a detailed overview of the mutation landscape of human CCA cell lines. Noticeably, the data presented here indicate that the genetic alterations occurring in CCA cells are also present in human CCA specimens, implying the potential use of these cell lines as a platform for drug testing. Nonetheless, we found that the response to targeted treatments of CCA cell lines harboring actionable mutations is heterogeneous, suggesting that the effectiveness of tailored therapies might depend on the intricated crosstalk between driver mutations and additional genetic and epigenetic events.

## Figures and Tables

**Figure 1 jpm-12-01270-f001:**
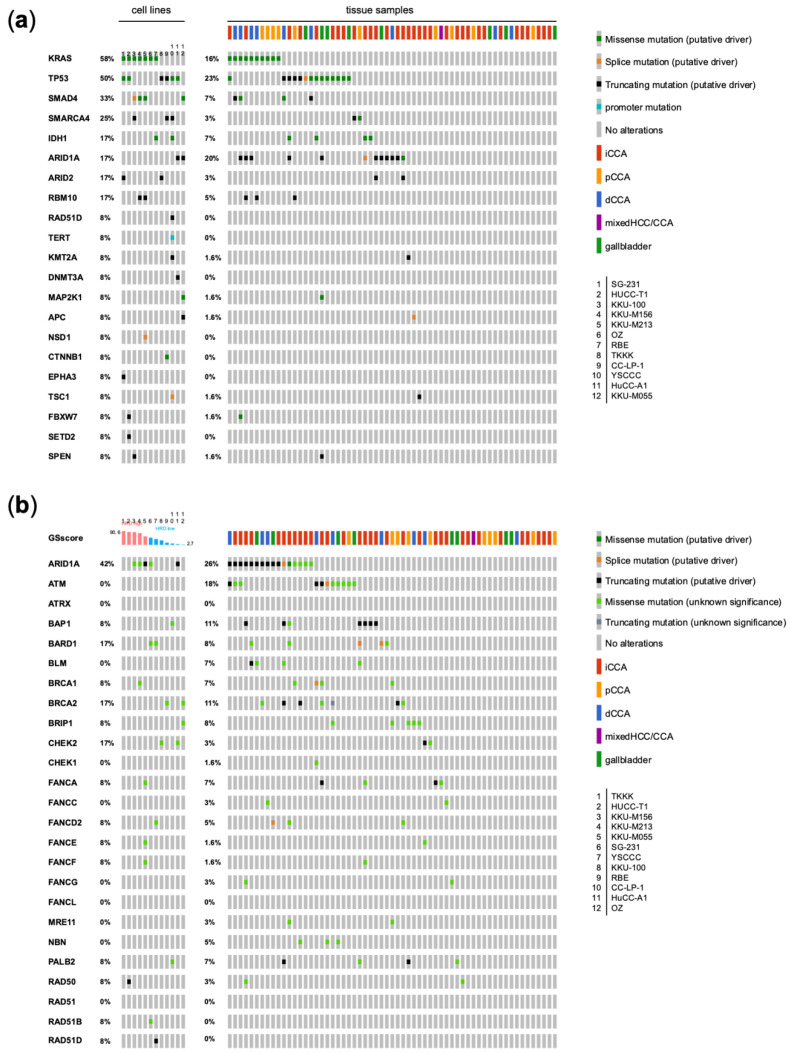
(**a**) Oncoprint plot displaying putative driver mutations detected in cell lines and CCA (cholangiocarcinoma) tissue samples with the frequency of occurrence. (**b**) Oncoprint plot including mutations of unknown significance in a set of homologous DNA repair associated genes. Indication of genomic scar score (GSscore) as determined by the AmoyDx^®^ HRD Focus Panel for cell lines. Abbreviations: iCCA, intrahepatic cholangiocarcinoma; pCCA, perihilar cholangiocarcinoma; dCCA, distal cholangiocarcinoma; mixed HCC/CCA, mixed hepatocellular cholangiocarcinoma; gallbladder, gallbladder carcinoma; HRD, homologous DNA repair deficiency.

**Figure 2 jpm-12-01270-f002:**
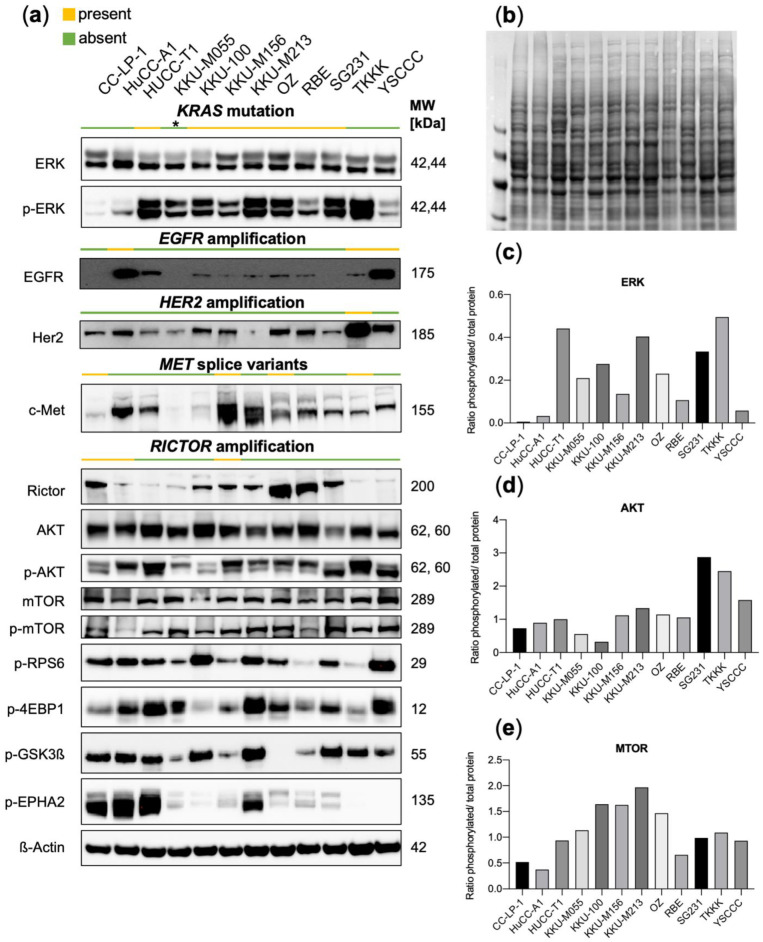
(**a**) Western Blot analyses of 12 CCA cell lines, including effectors of the RAS-MAPK and PI3K-MTOR pathways with the indication of corresponding alteration status. Asterisk indicates that KKUM055 harbors another MAPK-pathway alteration, which is *MAP2K1* (p.K57T). (**b**) Ponceau S staining of a representative membrane showing equal protein loading as in (**a**). (**c**) Quantification of band intensity ratios [phosphorylated (p)/ total protein levels] for ERK, (**d**) AKT, and (**e**) MTOR. No technical or experimental repeats of Western Blots were performed (*n* = 1). MW, molecular weight.

**Figure 3 jpm-12-01270-f003:**
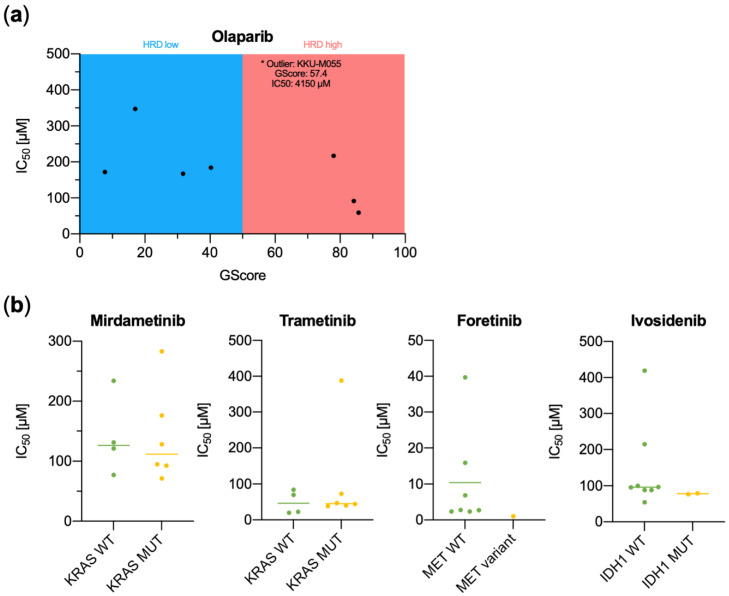
IC_50_ values as determined by Xcelligence^®^ gradient concentration measurements for the compounds (**a**) Olaparib displayed against genomic scar score (GScore), (**b**) Mirdametinib and Trametinib with regard to *KRAS* mutational status, Foretinib with regard to detected *MET* splice site variants and gene fusions, and Ivosidenib sorted by *IDH1* mutational status. HRD, homologous DNA repair deficiency; WT, wild type; MUT, mutated.

**Figure 4 jpm-12-01270-f004:**
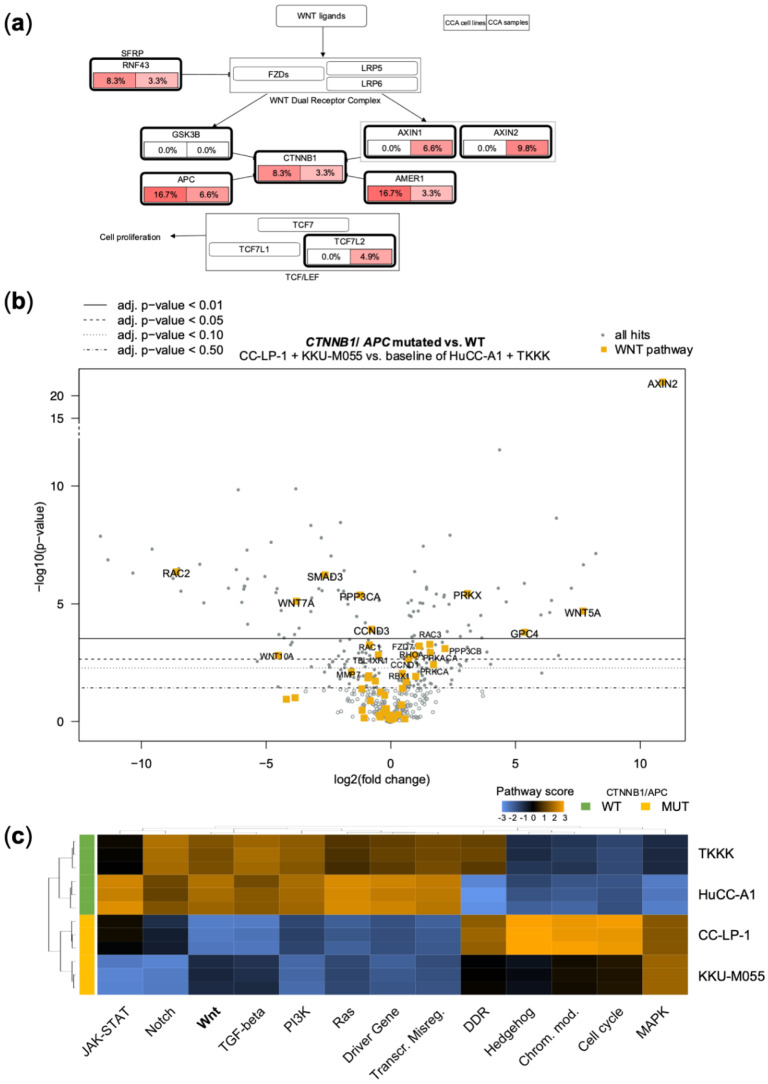
(**a**) Map of WNT pathway-associated genes. The mutational frequencies, including putative driver mutations and variants of unknown significance, are indicated for cell lines (*left*) and CCA tissue samples (*right*). (**b**) Volcano plot of Nanostring^®^ mRNA measurement using the human Pan Cancer pathway panel^®^ with highlighted Kegg WNT-pathway hits. Comparison of the *CTNNB1* mutated CC-LP-1 and the *APC* mutated KKU-M055 cell line with the two WNT pathway wild-type cell lines HuCC-A1 and TKKK. (**c**) Heatmap showing pathway scores across indicated cholangiocarcinoma cell lines to illustrate clustering of pathways within samples. Orange indicates high pathway scores; blue indicates low pathway scores (with undetermined functional dimension).

**Figure 5 jpm-12-01270-f005:**
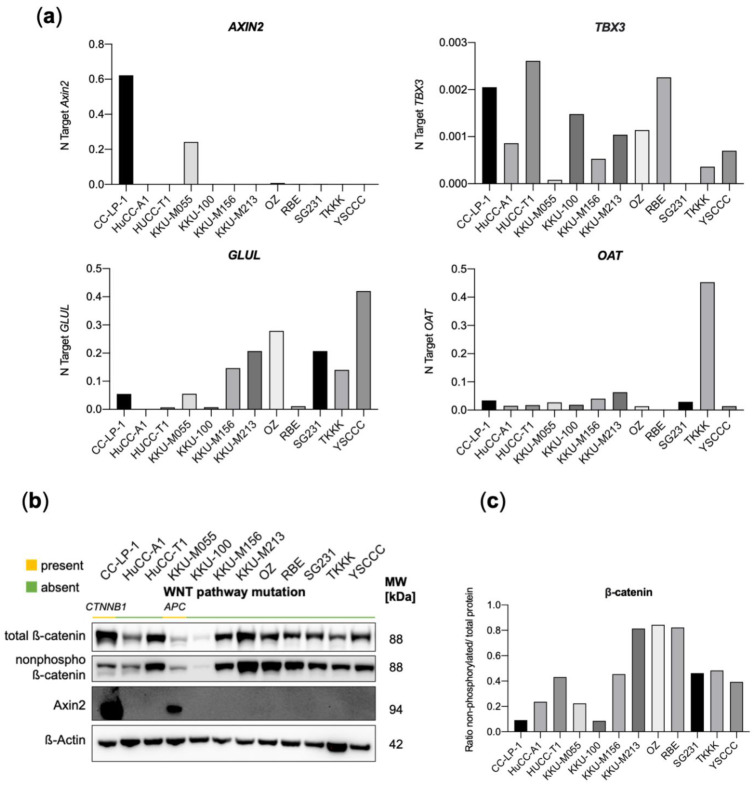
(**a**) Quantitative real-time PCR analyses of *Axin2*, *TBX3*, *GLUL*, and *OAT* mRNA levels in CCA cell lines. Quantitative values are expressed as number target (N Target). N Target = 2^−^^ΔCt^, wherein the ΔCt value of each sample was calculated by subtracting the average Ct value of the gene of interest from the average Ct value of the *GAPDH* gene. (**b**) Western Blot analysis of total β-catenin, non-phospho β-catenin, and Axin2 with (**c**) respective band volume intensity ratios. No technical or experimental repeats of Western Blots were performed (*n* = 1). MW, molecular weight.

**Figure 6 jpm-12-01270-f006:**
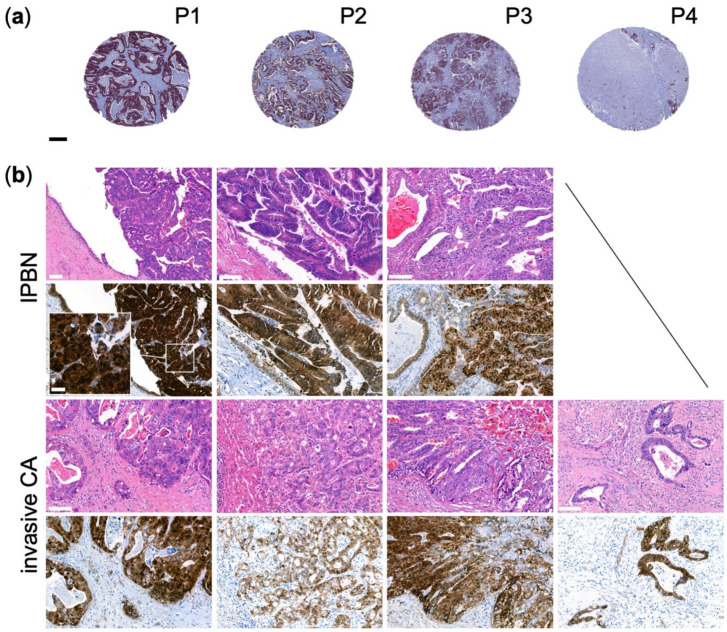
(**a**) Four Cholangiocarcinoma samples (out of 176 analyzed in a tissue microarray) with nuclear β-catenin staining detected by immunohistochemistry with validation in (**b**) primary liver resection specimen paraffin-embedded tissue blocks. P1–P3 with corresponding precursor lesions (Intraductal papillary neoplasm of the bile duct, IPNB). H&E (top rows) with respective β-catenin immunohistochemistry (bottom rows). Scale bar in (**a**) 250 µm in (**b**) 100 µm, except for inset 20 µm. P, patient identifier; CA, carcinoma.

**Figure 7 jpm-12-01270-f007:**
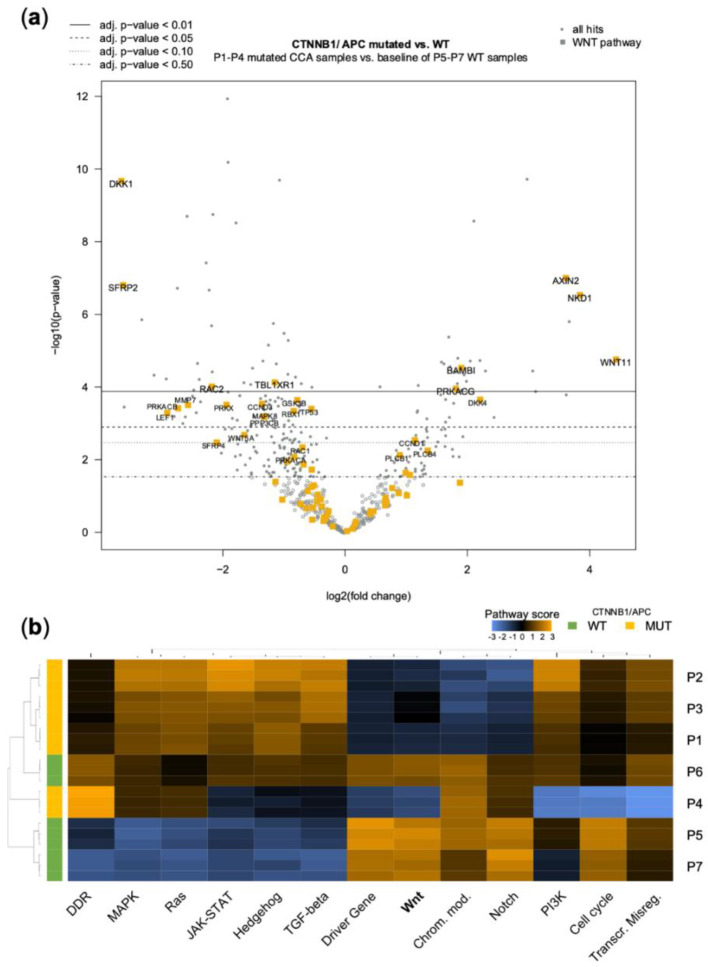
(**a**) Volcano plot of Nanostring^®^ mRNA measurement using the human Pan Cancer pathway panel^®^ with highlighted Kegg WNT-pathway hits. Comparison of the *CTNNB1* mutated CCA (P4) and the *APC* mutated CCA samples (P1–P3) with the three WNT pathway wild-type CCA samples (P5–P7). (**b**) Heatmap showing pathway scores across indicated patients to illustrate clustering of pathways within samples. Orange indicates high pathway scores; blue indicates low scores (with undetermined/arbitrary functional dimension). P, patient identifier.

**Table 1 jpm-12-01270-t001:** List of antibodies used for Western Blotting and immunohistochemistry (IHC).

Antigen [Clone if Monoclonal Antibody]	Dilution	Company	Catalog
Axin2 [H-98]	1/1000	Santa Cruz Biotechnology, Inc.	sc-14029
c-Met [EPR19067]	1/1000	Abcam plc.	ab216574
EGFR [EP38Y]	1/1000	Abcam plc.	ab52894
HER2/ErbB2	1/1000	Cell Signaling Technology, Inc.	2242
mTOR	1/1000	Merck Millipore	04-385
Non-phospho (Active) β-Catenin (Ser45) [D2U8Y]	1/1000	Cell Signaling Technology, Inc.	19807
Phospho-Akt (S473)	1/1000	Cell Signaling Technology, Inc.	4060
Phospho-mTOR (S2448) [49F9]	1/1000	Cell Signaling Technology, Inc.	2976
Phospho-44/42 MAPK (Erk1/2) [137F5]	1/1000	Cell Signaling Technology, Inc.	4695
Phospho-4EBP1 [53H11]	1/1000	Cell Signaling Technology, Inc.	9644S
Pan-AKT [C67E7]	1/1000	Cell Signaling Technology, Inc.	4691
Phospho-EPHA2	1/1000	Santa Cruz Biotechnology, Inc.	sc-924
Phospho-GSK3ß (S9) [D85E12]	1/1000	Cell Signaling Technology, Inc.	5558
Phospho-p44/42 MAPK (Erk1/2) (Thr202/Tyr204) [D13.14.4E]	1/1000	Cell Signaling Technology, Inc.	4370
Phospho-RPS6 (S235/236) [2F9]	1/1000	Cell Signaling Technology, Inc.	4856
Rictor	1/1000	Bethyl Laboratories	A300-459A
ß-catenin (used for IHC)	1/100	BD Biosciences	610154
ß-catenin (used for IHC)	1/100	Cell Signaling Technology, Inc.	9587
ß-catenin (used for IHC)	1/100	Roche	760-4242
β-Actin [8H10D10]	1/1000	Cell Signaling Technology, Inc.	3700
HRP, Goat Anti-Mouse IgG (secondary antibody)	1/20000	Abbkine Scientific Co., Ltd.	A21010
HRP, Goat Anti-Rabbit IgG (secondary antibody)	1/20000	Abbkine Scientific Co.	A21020

**Table 2 jpm-12-01270-t002:** List of TaqMan^®^ probes used for quantitative real-time PCR.

Gene	Assay ID	Fluorescent Dye	Manufacturer
*Axin2*	Hs00610344_m1	FAM	Thermo Fisher Scientific
*TBX3*	Hs00195612_m1	FAM	Thermo Fisher Scientific
*GLUL*	Hs00365928_g1	FAM	Thermo Fisher Scientific
*OAT*	Hs_00236852_m1	FAM	Thermo Fisher Scientific
*GAPDH*	Hs_02786624_g1	VIC	Thermo Fisher Scientific

**Table 3 jpm-12-01270-t003:** Overview of the genetic profiling of CCA cell lines. Qiagen Clinical Insight^®^ annotation.

	OZ	RBE	TKKK	YSCCC	HuCC-A1	HuCC-T1	KKU-M055	KKU-100	KKU-M156	KKU-M213	CC-LP-1	SG231
**GScore**	2.7	17	90.6	40.3	7.7	85.7	57.4	31.7	84.2	78	9.2	47.3
**TMB [mutations/MB]**	2.5	2.4	5.5	34.6	9.7	6.3	6.3	3.1	7.8	46.1	0.9	6.3
**Percent unstable MSI Sites**	5.5	3.61	0.91	4.44	0	6.41	0	0.91	7.08	4.35	8.33	1.69
**CNV** **>2 fold change**	none	none	EGFR (2.535) ERBB2 (10.840)	EGFR (13.822)	RICTOR (3.168) FGF10 (2.585) EGFR (2.310)	none	none	ERCC2 (2.133) AR (2.197)	FGF5 (5.452) RICTOR (3.926) ATM (2.617)	NRAS (2.001) FGF5 (3.158)	RICTOR (2.054)	MYC (3.694)
**Fusions**											MET-CAV1	EWSR1-ADRBK2
**Mutations (pathogenic)**	KRAS (p.Q61L) GNAS (p.R201H)	IDH1 (p.R132S) KRAS (p.G12V)	TP53 (p.Q104*)	RAD51D (p.K111fs*13) PIK3CA (p.E545K) TP53 (p.V272M) TERT (c.-124C>T)		KRAS (p.G12D) TP53 (p.R175H)	SMAD4 (p.R361H)	KRAS (p.G12D)	KRAS (p.G13C) SMAD4 (p.R361C)	KRAS (p.G13C) SMAD4 (p.R361C)	TP53 (p.R306*) CTNNB1 (p.G34V)	KRAS (p.G12F) TP53 (p.T125P) ARID2 (p.Q1041*) EPHA3 (p.C202*)
												MET (p.L130L)
**Mutations (likely pathogenic)**			ARID2 (p.K569fs*12)	IDH1 (p.R100Q) KMT2A (p.S82*) SMARCA4 (p.E457*) TSC1 (c.107-1G>A)	DNMT3A (p.R143*) TP53 (p.G154V) ARID1A (p.S1138*)	TERT (c.57A>C) FBXW7 (p.S294*) SETD2 (p.Q2285*)	ARID1A (p.P946fs*20) MAP2K1 (p.K57T) APC (p.D1636fs*2) RPS6KA4 (p.Q129*)	SMARCA4 (p.K1412*) SMAD4 (c.1140-1G>T) SPEN (p.Q743*)	RBM10 (p.E670*)	AR (p.L627Q) HRAS (p.V29V) NCOA3 (c.2513-1G>T) NSD1 (c.3115-1G>T) RBM10 (p.E670*)	SMARCA4 (p.E590*)	

**Table 4 jpm-12-01270-t004:** IC_50_-values [M] of 10 evaluable CCA cell lines. ND, not determined.

	RBE	YSCCC	HuCC-A1	HuCC-T1	KKU-M055	KKU-100	KKU-M156	KKU-M213	CC-LP-1	SG231
**Mirdametinib**	9.49 × 10^5^	2.34 × 10^4^	1.21 × 10^4^	9.27 × 10^5^	7.71 × 10^5^	1.76 × 10^4^	7.13 × 10^5^	2.83 × 10^4^	1.31 × 10^4^	1.28 × 10^4^
**Trametinib**	4.68 × 10^5^	6.99 × 10^5^	8.38 × 10^5^	4.42 × 10^5^	2.00 × 10^5^	7.26 × 10^5^	3.85 × 10^5^	3.88 × 10^4^	2.27 × 10^5^	4.00 × 10^5^
**Foretinib**	2.72 × 10^6^	2.37 × 10^6^	6.84 × 10^6^	ND	3.97 × 10^5^	1.59 × 10^5^	ND	2.77 × 10^6^	1.05 × 10^6^	2.38 × 10^6^
**Olaparib**	3.47 × 10^4^	1.84 × 10^4^	1.72 × 10^4^	5.90 × 10^5^	4.15 × 10^3^	1.67 × 10^4^	9.16 × 10^5^	2.17 × 10^4^	1.78 × 10^4^	1.03 × 10^4^
**Ivosidenib**	7.93 × 10^5^	7.67 × 10^5^	9.68 × 10^5^	5.45 × 10^5^	8.84 × 10^5^	8.80 × 10^5^	2.15 × 10^4^	9.54 × 10^5^	4.19 × 10^4^	9.96 × 10^5^

## Data Availability

Nanostring^®^ gene expression raw data are available in the Gene Expression Omnibus (GEO) database repository. This data can be found here: https://www.ncbi.nlm.nih.gov/geo/query/acc.cgi/GSE203639 (last accessed 28 July 2022). The CCA sequencing data presented in this study are available on request from the corresponding author. These data are not publicly available due to ethical restrictions.

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
