# Peer review of "Wnt/β-Catenin-Pathway Alterations and Homologous Recombination Deficiency in Cholangiocarcinoma Cell Lines and Clinical Samples: Towards Specific Vulnerabilities"

_jpm, 2022, doi:10.3390/jpm12081270_

Round 1

Reviewer 1 Report

Dear Editor,

The manuscript Wnt/b-catenin-pathway alterations and homologous recombination deficiency in cholangiocarcinoma cell lines and clinical samples: towards specific vulnerabilities interestingly reports the establishment of molecular profiling of CCA cell lines in comparison with CCA patients’ tumor samples, focusing on HRD-related genes and investigating CCA cell line signaling pathways and sensitivity to PARP inhibitor Olaparib and to other compounds, for personalized medicine purpose.

I have found the article justified for this journal based on application-orientated research journal devoted to reporting advances with originality and novelty, in the science and technology of biological research related to personalized medicine.

The outcome of research is interesting; however, the manuscript cannot be considered for publication as it stands. The study contains number of scientific flaws in terms of accuracy of information, data presentation and in the clarity.

1.      In the Introduction, the authors should clarify alterations in “individual alterations” (l52). The terms “DNA-damage related genes” should be also revised and completed. The acronym HRD is sometimes typed as HR or as HRR (in the Results section and supplementary Table), which should be corrected as well. The removal of the hyphen from “Wnt-ligands” is suggested.

2.      In the Materials and Methods section, the authors should revise the flow of the methodologies (e.g., Use of cell pellets firstly mentioned in Western blot, first methodology described – whereas cell information is given in the last methodology). Minutes (or min) and seconds (s) should be consistent. The full name of TBST and the use of the Western blot imaging system should be given. The name and brand of the DNA extraction kit used from CCA cell lines should be mentioned. The full name of Kegg should be given as well. The authors are advised to define iCCA, pCCA, dCCA, and eCCA. Correction on “100 mg/mL streptomycin” and on “100 U/mL penicillin”. Liter (L or l) should be revised for consistency as well. In the cell viability assay section, the known inhibitory function of each cytotoxic compound used should be mentioned (e.g., PARP inhibitor Olaparib).

3.      In the Results section, the authors should replace “lysates” with extracts and should add a hyphen in “TSO500® based” (l252-253). Once the abbreviations/acronyms are defined, then keep using the abbreviation/acronym only (e.g., CCA - l261, TMB - l329, CNV - l350). Recommendation of the addition of the term cells after each cell line name in order to avoid any confusion or terms related to gene or signaling protein. Regarding Western blot analysis, there are too many variations of expression level of total signaling protein such as ERK, AKT and mTOR, expected to show equal loading like the housekeeping protein b-actin. The authors should provide better Western blot results regarding these aforementioned proteins. If not, can they explain why do we see these variations of total ERK, AKT and mTOR protein expression levels from these different cell lines knowing that the loaded protein amount is fixed and must be equal between each sample? Addition of the molecular weight should be indicated along each Western blot image. Concerning the Western blot analysis for mTOR, the ratio value is shown to reach around 0.3-0.6 although the related bands corresponding to SG231, TKKK and YSCCC cells are not clearly seen. Can the authors show more relevant mTOR Western blot results corresponding to these related Western blot analysis? Addition of error bars and statistical analysis are also recommended with indication of the number of independent experiments carried out. For consistency, correct the y-axis of the mTOR bar graph. P for P1 and P2 (lines 502-504) should be defined (before reaching Figure legends). For consistency, correct HuCC-T1 and SG231 above the Western blot images and legends of bar graphs.

4.      Overall the Discussion is well written. However, few corrections should be made: addition of were to “BAP1 found” (l540). DNA-damage related (DDR) genes must be revised and corrected as aforementioned.

I hope that my comments will contribute to the improvement of the manuscript and will help you make a decision on the quality of this manuscript.

Faithfully yours,

Author Response

I have found the article justified for this journal based on application-orientated research journal devoted to reporting advances with originality and novelty, in the science and technology of biological research related to personalized medicine.

  • We thank the reviewer for this positive assessment of our work.

The outcome of research is interesting; however, the manuscript cannot be considered for publication as it stands. The study contains number of scientific flaws in terms of accuracy of information, data presentation and in the clarity.

  1. In the Introduction, the authors should clarify alterations in "individual alterations" (l52). The terms "DNA-damage related genes" should be also revised and completed. The acronym HRD is sometimes typed as HR or as HRR (in the Results section and supplementary Table), which should be corrected as well. The removal of the hyphen from "Wnt-ligands" is suggested.
  • The definition "Individual alterations" was replaced by "targetable molecular alterations".
    The term "DNA-damage related genes" was replaced by homologous recombinational repair (HRR) related genes. As the reviewer correctly noted, the acronym HRR had not been defined in the initial version of our manuscript. HRR is different from HRD (HRD means the deficiency in homologous recombinational repair). The term HRR-related genes is now used consistently throughout the revised manuscript. The hyphen in Wnt ligands was removed according to the reviewer's suggestion.
  1. In the Materials and Methods section, the authors should revise the flow of the methodologies (e.g., Use of cell pellets firstly mentioned in Western blot, first methodology described – whereas cell information is given in the last methodology). Minutes (or min) and seconds (s) should be consistent. The full name of TBST and the use of the Western blot imaging system should be given. The name and brand of the DNA extraction kit used from CCA cell lines should be mentioned. The full name of Kegg should be given as well. The authors are advised to define iCCA, pCCA, dCCA, and eCCA. Correction on "100 mg/mL streptomycin" and on "100 U/mL penicillin". Liter (L or l) should be revised for consistency as well. In the cell viability assay section, the known inhibitory function of each cytotoxic compound used should be mentioned (e.g., PARP inhibitor Olaparib).
  • The authors fully agree on the inconsistency of the flow of methodologies. "Cell culture and in vitro analyses" has been changed to 2.1 instead of 2.9. Minutes was replaced by min, seconds by s and hours by h throughout the manuscript.
    TBST was specified: Tris Buffered Saline with Tween® 20 (Cell Signaling Technology, Inc., Cambridge, UK). 
    Image acquisition was performed on a ChemiDocTM MP Imaging System (Bio-Rad Laboratories).
    The AllPrep® DNA Mini Kit (Qiagen Sciences Inc., Germantown, USA) was used for DNA extraction from CCA cell lines.
    Kegg was spelled out as Kyoto Encyclopedia of Genes and Genomes.
    The abbreviations iCCA, pCCA, dCCA, and eCCA were spelled out. An explanation is given for eCCA, which comprises both pCCA and dCCA.
    Liter (l) was consistently written.
    As the reviewer recommended, the inhibitory functions of the used cytotoxic compounds were added in the methods section.
  1. In the Results section, the authors should replace "lysates" with extracts and should add a hyphen in "TSO500® based" (l252-253). Once the abbreviations/acronyms are defined, then keep using the abbreviation/acronym only (e.g., CCA - l261, TMB - l329, CNV - l350). Recommendation of the addition of the term cells after each cell line name in order to avoid any confusion or terms related to gene or signaling protein. Regarding Western blot analysis, there are too many variations of expression level of total signaling protein such as ERK, AKT and mTOR, expected to show equal loading like the housekeeping protein b-actin. The authors should provide better Western blot results regarding these aforementioned proteins. If not, can they explain why do we see these variations of total ERK, AKT and mTOR protein expression levels from these different cell lines knowing that the loaded protein amount is fixed and must be equal between each sample? Addition of the molecular weight should be indicated along each Western blot image. Concerning the Western blot analysis for mTOR, the ratio value is shown to reach around 0.3-0.6 although the related bands corresponding to SG231, TKKK and YSCCC cells are not clearly seen. Can the authors show more relevant mTOR Western blot results corresponding to these related Western blot analysis? Addition of error bars and statistical analysis are also recommended with indication of the number of independent experiments carried out. For consistency, correct the y-axis of the mTOR bar graph. P for P1 and P2 (lines 502-504) should be defined (before reaching Figure legends). For consistency, correct HuCC-T1 and SG231 above the Western blot images and legends of bar graphs.
  • The minor corrections were applied. Abbreviations were used more stringently as suggested. The proposal to add the term cells after the scientific nomenclature was followed.
    We especially thank the reviewer for pointing out the unusual variation in ERK, AKT, and mTOR expression levels. Given the reviewer's insightful input, we could deduce that the bands' inconsistency was a technical issue. Therefore, the respective Western Blots were repeated. By using wet blotting (tank blots) instead of dry blotting, it was possible to achieve more homogenous results, particularly for the large protein mTOR. Furthermore, we added a Ponceau stain, which indicates homogenous protein loading in addition to the β-actin bands. Observed molecular weights are now indicated on the right of the included Western blots. Western Blots were only carried out once in the respective conditions (also due to time constraints for the revisions). Nevertheless, the Ponceau Red staining clearly confirmed equal loading. This limitation is now clearly indicated in the figure captions. P is an acronym for patient and has now been defined.
  1. Overall, the Discussion is well written. However, few corrections should be made: addition of were to "BAP1 found" (l540). DNA-damage related (DDR) genes must be revised and corrected as aforementioned.
  • The corrections were made as suggested. Thank you for pointing these mistakes out.

Reviewer 2 Report

The study focused on the molecular profile of CCA cell lines and comparison with patient's tumor samples to design new targeted therapeutic strategies.

The study on mechanisms of cholangiocarcinogenesis, conducted by several new data processing procedures, stimulates further investigations to discover new elements otherwise obscure.

The procedures utilised are fruitful also to obtain informations in CCA prognosis, about mutation panel correlated with previous malignant neoplastic disease (colorectal cancers). 

Author Response

The study focused on the molecular profile of CCA cell lines and comparison with patient's tumor samples to design new targeted therapeutic strategies.

The study on mechanisms of cholangiocarcinogenesis, conducted by several new data processing procedures, stimulates further investigations to discover new elements otherwise obscure.

The procedures utilised are fruitful also to obtain information on CCA prognosis, about mutation panel correlated with previous malignant neoplastic disease (colorectal cancers).

  • We thank the reviewer for this positive assessment of our work and figure that no changes are warranted on these grounds.

Round 2

Reviewer 1 Report

Dear Editor,

The authors have considered my comments and improved the quality of the manuscript. 
Their manuscript can be accepted after correction of minor mistakes still left. For instance: i. The abstract, full name of HRR before use of abbreviations. Correction of streptomycin concentration (100 microg/ml and not my/ml), etc..